Impact of atmospheric particulate matter retention on physiological characters of five plant species under different pollution levels in Zhengzhou

He Dan 1
Yuan Jiangqin 1
Lin Runze 2
Xie Dongbo 3
Wang Yifei 1
Kim Gunwoo 4
Lei Yakai 1 lykfjyl@163.com
Li Yonghua 1 liyhhany@163.com
1 College of Landscape Architecture and Art, Henan Agricultural University , Zhengzhou, Henan , China
2 Landscape Architecture Department, Huazhong Agricultural University , Wuhan, Hubei , China
3 Research Institute of Forest Resource Information Techniques, Chinese Academy of Forestry , Beijing , China
4 Graduate School of Urban Studies, Hanyang University , Seoul , Republic of South Korea
Sadeghi Seyed Mohammad Moein
Electronic publication date: 2024 Sep 27
Publication date: 2024
Volume: 12
Electronic Location ID: e18119
Received 2024 May 16; Accepted 2024 Aug 28
Copyright: © 2024 He et al.
Copyright year: 2024
Copyright holder: He et al.
License: This is an open access article distributed under the terms of the Creative Commons Attribution License, which permits unrestricted use, distribution, reproduction and adaptation in any medium and for any purpose provided that it is properly attributed. For attribution, the original author(s), title, publication source (PeerJ) and either DOI or URL of the article must be cited.
License URL: https://creativecommons.org/licenses/by/4.0/

Keywords: Landscape plants, Air pollution, Particulate matter, Dust retention, Physiological response

Funding: National Natural Science Foundation of China 31600579 Henan Provincial Science and Technology Research 212102110185 Key Young Teachers of Henan Province, China 2020GGJS049 This research was supported by the National Natural Science Foundation of China [31600579], the Henan Provincial Science and Technology Research project [212102110185], and the Key Young Teachers of Henan Province, China [2020GGJS049]. The funders had no role in study design, data collection and analysis, decision to publish, or preparation of the manuscript.

==============================
Atmospheric particulate matter (PM) pollution has become a major environmental risk, and green plants can mitigate air pollution by regulating their enzymatic activity, osmoregulatory substances, photosynthetic pigments, and other biochemical characteristics. The present investigation aims to evaluate the mitigation potential of five common evergreen tree species (Photinia serrulata, Ligustrum lucidum, Eriobotrya japonica, Euonymus japonicus, Pittosporum tobira) against air pollution and to assess the effect of dust retention on plant physiological functions exposed to three different pollution levels (road, campus, and park). The results found that the amount of dust retained per unit leaf area of the plants was proportional to the mass concentration of atmospheric particulate matter in the environment, and that dust accumulation was higher on the road and campus than in the park. There were significant differences in dust retention among the five tree species, with the highest leaf dust accumulation observed for E. japonica (5.45 g·m−2), and the lowest for P. tobira (1.53 g·m−2). In addition, the increase in PM adsorption by different plants was uneven with increasing pollution levels, with significant decreases in chlorophyll content, photosynthetic and transpiration rate. From a physiological perspective, P. tobira exhibited greater potential to respond to PM pollution. Biochemical indicators suggested that PM pollution caused changes in plant protective enzyme activities, with a decrease in superoxide dismutase (SOD) and peroxidase (POD) activities, as well as promoting membrane lipid peroxidation, and appropriate stress also enables plants to counteract oxidative damage. In particular, PM exposure also induced stomatal constriction. Overall, PM retention was significantly associated with physiological and photosynthetic traits. In conclusion, our study contributes to the understanding of the effects of PM on plant physiology. Furthermore, it also provides insights into the selection of plants that are tolerant to PM pollution.

Introduction

The rapid advancement of industrialisation and urbanisation has taken a toll on the Earth’s ecology and atmosphere, with one of the most pressing environmental concerns being atmospheric particulate matter (PM) pollution (Zhang et al., 2019; Zhou et al., 2020), which has resulted in negative impacts on human health and biodiversity in urban areas (Khalid et al., 2019). According to a report by the State of the Chinese Environment in 2021, 35.7% of the 339 cities at the prefecture level and above were found to have exceeded the air quality standard. PM, which can carry heavy metals, bacteria, viruses, and other harmful substances, poses a significant threat to human health, and is considered the primary risk factor for global mortality (Konczak et al., 2023; Dang et al., 2022; Yu et al., 2023). The smaller the particle size, the more harmful it is to the human body. PM10 (Dp ≤ 10 μm) and PM2.5 (Dp ≤ 2.5 μm) cause potential threat to human health in particular (Maher et al., 2022). PM10 can lead to chronic rhinitis or bronchitis, while PM2.5 can adversely affect the cardiovascular system, increase the incidence of respiratory infections, and may even be life-threatening (Manigrasso et al., 2020; Yang, Li & Tang, 2020). Furthermore, PM can pose various hazards, including reducing atmospheric visibility, contributing to the greenhouse effect, and indirectly affecting the balance of terrestrial and marine ecosystems (Dong et al., 2016). Consequently, reducing the concentration of atmospheric PM in cities has become an urgent issue.

Planting vegetation is an effective ecological strategy to alleviate PM pollution and improve urban air quality (Weerakkody et al., 2017; Hewitt, Hashworth & MacKenzie, 2019; Yazbeck et al., 2021). According to a study in Canada, 86 urban forests removed 16,500 tons of air pollution in 2010 (Nowak et al., 2018). Additionally, Parsa et al. (2019) emphasized that urban plants can capture 814.46 tons of atmospheric pollutants over the next 20 years with reasonable planting management. Currently, researches are primarily focused on comparing the dust retention capacity of different plants and investigating the influence of leaf microstructure characteristics on PM retention (Zhang et al., 2018; Li et al., 2021). Previous studies have demonstrated that the microstructure present on the leaf surface plays a significant role in PM retention. Leaves with fluffy groove structures, trichomes, high stomatal density, and thick wax layers show greater capacity for accumulating PM (Zhang et al., 2018, 2021; Sæbø et al., 2012), while plants with smooth blade surfaces had poor adsorption capacity. Based on previous researches, scholars have generally proposed suggestions for the selection of tree species, while neglecting the damage to plant function (Li et al., 2023).

While mitigating air pollution, plants are also adversely affected by PM and are able to adapt to this stress by regulating various physiological and biochemical traits (Zhu & Xu, 2021; Singh et al., 2020). Previous studies found that the PM pollution can reduce the chlorophyll content in plant leaves, thereby affecting the synthesis of other pigments and derivatives, as well as hindering the synthesis of proteins (Shah et al., 2018; Singh et al., 2018, 2020). Exposure to PM pollution can disrupt the equilibrium between the production and removal of reactive oxygen species (ROS) in plant leaves, resulting in the accumulation of free radical, which leads to oxidative stress and ultimately cell death in plants (Singh et al., 2020; Turan, 2022). As this time, the antioxidant system in plant body detoxifies ROS in plants through different mechanisms such as superoxide dismutase (SOD) and peroxidase (POD) to reduce the harm of oxidative stress (Li et al., 2023). Furthermore, PM will form a dust layer on the leaves, causing stomatal blockage, which in turn affects the gas exchange between the leaves and the external environment. This in turn affects plant physiological processes, including photosynthesis and respiration (Selmia et al., 2016; Wuyts et al., 2018; Yan et al., 2018). In a study conducted by Sabir et al. (2023), it was found that dust had negative impact on the stomatal conductance, photosynthesis, and transpiration rate of urban tree species in semiarid regions. By triggering the activities of SOD and POD, Bombax ceiba effectively eliminated ROS and demonstrated stronger tolerance to dust pollution. Thus, understanding the response of urban vegetation to PM pollution is crucial for selecting appropriate tree species for urban greening.

In addition to the characteristics of the plants themselves, the dust retention of various tree species is also influenced by regional environmental conditions (Jeanjean, Monks & Leigh, 2016). Plants in more polluted areas tend to absorb more dust compared to those in less polluted areas (Zhu & Xu, 2021; Dang et al., 2022). Currently, there is limited research on the physiological responses of plants to urban atmospheric PM stress under different pollution levels. The potential of many tree species to reduce PM pollution and their physiological responses under different conditions remain unclear. Zhengzhou, located in the central region of China, is a city facing challenges associated with rapid urbanisation and climate change. This study aimed to investigate the capacity of five typical tree species to adsorb PM in three polluted areas and to examine their physiological responses. It is assumed that the physiological functions of different plants are self-regulating in response to different levels of particulate pollution. With this hypothesis, the present study was conducted with the following objectives: (1) to evaluate the foliar dust retention effect among different plant species in different polluted areas; (2) to analyse the specific changes in physiological and biochemical indicators of tree species; and (3) to identify tree species with strong dust retention and anti-pollution capacity. The research results will help to understand the relationship between PM pollution and plant physiology, and provide reference and basis for the construction and management of urban greening tree species.

Materials and Methods

Research area description

Zhengzhou, located in the center of China, has a continental monsoon climate in northern temperate zone, with four distinctive seasons. The average annual temperature ranges from 12–22 °C, the average wind is 2–2.8 m·s−1, and the average annual rainfall is 640.9 mm. The texture of the soil is loamy clay.

The sampling sites were selected in Zhongzhou Avenue, Henan Agricultural University, and the Botanical Garden in Zhengzhou, China (refer to Fig. 1), and the sampling was approved by Zhengzhou Botanical Garden. The selection of these three sites was based on factors such as air pollution levels, transportation conditions, and crowd flow. Zhongzhou Avenue, an important section of the Zhengzhou Expressway, is 2.1 km long and 7 m wide. Traffic is heavy, with an approximate flow of 45 vehicles per minute (two-way). Henan Agricultural University is situated at No. 63, Nongye Road, Jinshui District, and is surrounded by schools and residential areas in an old urban area. It exhibits moderate pollution level compared to the other two sampling areas. Zhengzhou Botanical Garden is located in the suburbs along the West Fourth Ring Road. This area is designated as a clean control area with no large industries or heavy pollution sources nearby. The three sites above represent different levels of PM pollution, and the sites were classified as high, medium, and low (control) areas on the basis of the airborne total suspended particulate matter (TSP) concentrations measured at each site, which were 274.2, 191.1, and 80.4 μg·m−3 respectively.

Figure 1 Distribution of sampling area.

Source: Bigemap Gis Software. Map data: Google.

Plant species and sampling collection

According to the primary survey of greening trees in all three sampling sites, five evergreen tree species, including Photinia serrulata, Euonymus japonicus, Pittosporum tobira, Ligustrum lucidum, and Eriobotrya japonica were selected. These species are widely used in northern China, and are also the representative plant species in Zhengzhou City. The details of the tested plants are described in Table 1.

Table 1 Description of experimental trees species.

Plant species	Family	Life form	Leaf morphology characteristics	
Photinia serrulata	Rosaceae	Arbor	Leaves are long oval and leathery, with finely serrated edges	
Ligustrum lucidum	Oleaceae	Arbor	Both sides of leaf blade are glabrous	
Eriobotrya japonica	Rosaceae	Arbor	Leaves are lanceolate, with rough and wrinkled surface	
Euonymus japonicus	Buxaceae	Shrub	Leaves are thin and leathery, oval in shape, with relatively smooth surface	
Pittosporum tobira	Pittosporaceae	Shrub	Leaves are leathery, with smooth leaf surface and prominent midrib	

Based on the climatic conditions and rainfall characteristics of Zhengzhou City, the experiment was conducted in September 2020. To ensure saturation of PM on plant leaves, samples were collected approximately 14 days after rainfall. Three healthy plants of similar age with luxuriant foliage and free from pests and diseases were randomly selected from each plant in each region. For each tree species, the canopy was divided into three layers (lower, middle and upper) and further subdivided into four directions. A total of twelve collections (three layers and four directions per canopy) were obtained from a single tree. Leaf acquisition was facilitated using a flat-topped bifurcated aluminum ladder to access all strata and directions. Each sampling was aimed at an area of 200−300 cm2 of leaves, corresponding to 10–15 pieces for larger leaves and 20–30 pieces for smaller leaves, each species in triplicate. To avoid dehydration and structural changes, all samples were divided into two parts. The first part was stored at 4 °C for dust retention experiments and stomatal observation, while the second part was treated with liquid nitrogen and stored in an ultra-low temperature freezer at −80 °C (Forma905, Waltham, MA, USA) for physiological experiments.

Quantifying the PM retention on leaf surfaces

PM on leaves can be categorized into two parts: the wax layer and the leaf surface. However, the wax layer usually contains fewer particles, and the PM in the wax layer is easily souble in water, as mentioned by Dzierzanowski et al. (2011). The results of this study were only for the amount of physical, insoluble PM deposited on the leaf surface. The graded membrane filtration method was used to determine the amount of particulate matter adsorbed on the leaf surface per unit leaf area (Dzierzanowski et al., 2011). Initially, the collected leaves were soaked in distilled water for 2 h and gently cleaned with a soft brush. The leaves were then rinsed twice with approximately 200 mL of distilled water each time. The resulting suspensions from the three washes were combined and filtered using filter membranes with diameters of 10, 2.5, and 0.2 μm, respectively. The filtered membranes were then dried in a constant temperature drying oven set at 60 °C, and their mass was measured using a balance with a precision of 1/10,000. The particles retained on the filters were classified as PM>10, PM2.5–10, and PM2.5 based on their respective filter diameters. The total mass of TSP was calculated as the sum of the masses on all three filters, while the mass of PM10 was determined by adding the masses of PM2.5–10 and PM2.5. The leaf area of each plant was measured three times using a YMJ-B portable leaf area instrument (Topu Yunlong, Zhejiang Province, China). The PM retention capacity is calculated as the ratio of the amount of particles retained on the leaf surface to the leaf area.

Observation of stomata on the leaf surface

Mature leaves were carefully harvested and placed on glass microscope slides. Sections 1 cm × 1 cm were cut from each leaf, specifically from areas away from the midvein. To isolate the epidermis, both the upper epidermal layer and the mesophyll tissue were carefully removed during subsequent processing. Stomatal features such as size and number were then quantified using high-resolution imaging (1,200 x magnification) with an ultra-deep optical microscope (Leica DVM6A, Wetzlar, Germany).

Determination of leaf photosynthetic parameters

In September 2020, photosynthetic gas exchange parameters, including net photosynthetic rate (Pn), transpiration rate (E), stomatal conductance (Gs), intercellular CO2 concentration (Ci), were measured with a portable CIRAS-3 photosynthetic apparatus (USA) from 9:00 to 11:00 am after eight consecutive days without precipitation in sunny, windless or breezy weather.

Determination of leaf physiological indices

Chlorophyll a and b contents were determined by the ethanol extraction method, Weighted 0.1 g cut fresh leaves, add 10 ml of 95% ethanol until all the leaf tissue dissolved in the solution. After dark treatment for 24 h until the tissue turned white, the absorbance of the supernatant were determined at 649 nm (A649) and 665 nm (A665) using spectrophotometer (*UV-6100). Finally, calculated the chlorophyll content according to a relevant formulae (Li et al., 2023).

Malondialdehyde (MDA) content was quantified by the thiobarbituric acid method. Approximately 0.1 g of shredded fresh leaf tissue was ground in 1.5 ml of 5% trichloroacetic acid (TCA) solution, the extracted supernatant was mixed in equal proportions with 0.67% thiobarbituric acid solution (TBA) in a boiling water bath for 15 min, cooled rapidly and the supernatant was taken to measure the absorbance at 450, 532 and 600 nm, and the MDA content was subsequently calculated according to the relevant formulae (Li et al., 2023).

About 0.1 g fresh crushed leaves were placed in 5 ml centrifuge tubes with 2 ml of distilled water. The supernatant was extracted after grinding and centrifugation. Soluble sugar content was taken by the anthrone colorimetric method, and soluble protein content was determined via Coomassie brilliant blue-G250 staining. Enzyme activity was measured by adding 2 ml of phosphate buffer for extraction, the corresponding reaction mixture was then added to determine the superoxide dismutase (SOD) and peroxidase (POD) activity. It is important to note that the reaction mixture should be added in a certain order. SOD activity was measured by NBT photochemical reduction method (Beauchamp & Fridovich, 1971), and POD activity was determined via the guaiacol method (Zhao, Li & Gao, 2015).

After drying to constant weight at 70 °C, leaves were weighed using a balance with a precision of 1/10,000, and specific leaf weight (SLW) was calculated as the ratio of dry leaf weight to leaf area (g·m−2) (Ashiuchi et al., 2001).

Determination of fast chlorophyll fluorescence parameters

During the experiment, a random selection of 5–7 leaves was taken from inside and around the lower canopy of different experimental tree species that exhibited healthy growth conditions. The selected leaves were then underwent a 30-min dark adaptation, and their fast chlorophyll fluorescence parameters were determined using a multifunctional plant efficiency analyzer (M-PEA, Hansatech Instruments Ltd., Pentney, UK). The blade fully covered with a 4 mm2 test hole, while the measurement light source was a red light emitting a wavelength of 650 nm via six light emitting diodes, the light intensity used was 3,000 μmol·m−2·s−1. The fluorescence signal was recorded for a duration of 1 s, and three repeated measurements were taken for each tree species. The fast chlorophyll fluorescence curves provide several key fluorescence parameters, each of which possessed important physiological significance.

Tfm: Time to reach maximal fluorescence intensity Fm;

Fv/Fm = (Fm–Fo)/Fm: Maximum photochemical efficiency of PSII;

Vj: Relative variable fluorescence intensity at the J-step, representing the rate of energy dissipation of electrons as they pass through plastiquinone A(QA);

Vi: Relative variable fluorescence intensity in the I-step, representing the rate of energy dissipation of electrons as they pass through plastiquinone B (QB);

Sm: Normalised total complementary area above the O-J-I-P transie (reflecting single-turnover QA reduction events);

N: The number of times QA was restored during the period from the start of illumination to the arrival of Fm;

TR/RC: Trapped energy flux per RC (at t = 0);

ETo/RC: Electron transport flux per RC (at t = 0);

ΦRo: Reflects the relative activity of PSI;

PI(abs): Performance index on absorption basis.

Data analysis

A one-way analysis of variance (ANOVA) was conducted to analyse the differences in dust retention, leaf physiology, and stomatal characteristics among five tree species under different pollution sites. Subsequently, Pearson correlation analysis (PCA) was used to investigate the internal relationship of leaf physiological and biochemical characteristics and the associations between dust retention ability, leaf photosynthetic and physiological indicators. All statistical analyses were performed using SPSS 26.0 software (IBM., Armonk, NY, USA), with a significance level of 0.05. All charts were created using Excel 2010 (Microsoft Corp., Redmond, WA, USA) and Origin 2021 software (Origin Lab Corp., Northampton, MA, USA).

Results

The mass of PM retained on leaf surface in different pollution areas

There were significant differences in PM adsorption capacity among five tree species with different particle sizes (TSP, PM10 and PM2.5) (P < 0.05) (refer to Fig. 2). The dust retention of TSP, PM10 and PM2.5 ranged from 0.81 to 7.33, 0.20 to 3.68, and 0.12 to 3.60 g·m−2, respectively. Overall, E. japonica had the highest average TSP retention per unit leaf area (5.45 g·m−2), which was significantly higher than that of the other four species across the three sampling sites (P < 0.05), followed by E. japonicus (2.76 g·m−2) and P. serrulata (2.12 g·m−2), while L. lucidum (1.82 g·m−2) and P. tobira (1.53 g·m−2) showed the weakest dust retention ability. Similarly, it was found that E. japonica had the highest adsorption rate for PM10 and PM2.5, whereas P. tobira exhibited the weakest ability to capture PM10 and PM2.5.

Figure 2 Variation in dust retention per unit leaf area for five tree species under different pollution areas.

Road, Zhongzhou Avenue; Campus, Henan Agricultural University; Park, Zhengzhou Botanical Garden. Different lowercase letters indicate significant differences among different tree species in the same sampling area (P < 0.05).

There were variations in the adsorption of PM per unit leaf area under different pollution levels. Generally, as the pollution level increased, the plants showed greater adsorption of atmospheric PM. Among the three sampling areas, the order of TSP retention per unit leaf area for P. serrulata, P. tobira, L. lucidum, and E. japonica was Zhongzhou Avenue > Henan Agricultural University > Zhengzhou Botanical Garden. However, in the botanical garden, E. japonicus retained higher amount of TSP compared to the campus. The order of PM10 and PM2.5 retention for the five tree species generally followed the same pattern. In polluted areas, E. japonica adsorbed 7.8 times more PM2.5 per unit leaf area compared to the control area. Similarly, the PM2.5 retention of E. japonicus, P. serrulata, P. tobira, and L. lucidum on the road was 4.0, 3.1, 2.4, and 1.5 times higher, respectively, compared to the park.

Influence of atmospheric PM on the number and size of leaf surface stomata

The study examined the stomatal size and number in five different tree species. Upon magnification at 1,200 times, it was discovered that on campus, E. japonica had the longest stomatal length (34.7 μm), while in the botanical garden, P. tobira had the widest stomatal width (28.6 μm). The smallest stomatal length (17.5 μm) and width (13.6 μm) were observed in L. lucidum on the road, and the maximum number of stomata (53) was found in P. tobira on the road (refer to Fig. 3, Table 2). Furthermore, the study revealed that the stomatal characteristics of each species varied depending on the pollution levels. In all three sampling areas, P. serrulata, E. japonicus, and P. tobira exhibited downward trend in stomatal length and width as pollution levels increased. The largest stomatal area was observed in the botanical garden, while the smallest was found on the road.

Figure 3 Ultra-depth optical microscope observation of stomata on leaf surface of five plants under different pollution levels.

Table 2 Stomatal size and number of five plants leaves in three sampling areas.

Species	Sampling area	Stomatal length (μm)	Stomatal width (μm)	Stomatal number (piece)	
P. serrulata	Road	24.9 ± 1.31b	20.2 ± 0.79b	46 ± 4.36a	
Campus	28.8 ± 1.51a	24.3 ± 0.62a	44 ± 2.65a	
Park	29.9 ± 0.61a	25.1 ± 0.89a	41 ± 2.65a	
E. japonicus	Road	20.2 ± 0.7b	16.3 ± 1.06b	37 ± 5.57a	
Campus	21.6 ± 0.61ab	17.8 ± 1.11ab	39 ± 2.0a	
Park	23.3 ± 1.15a	19.1 ± 1.51a	38 ± 1.73a	
P. tobira	Road	30.4 ± 0.98b	26.5 ± 0.61b	53 ± 2.65a	
Campus	31.8 ± 0.72ab	27.3 ± 0.44b	29 ± 2.0c	
Park	32.7 ± 0.85a	28.6 ± 0.61a	41 ± 2.0b	
L. lucidum	Road	17.5 ± 0.79a	13.6 ± 0.7b	51 ± 2.65a	
Campus	18.7 ± 0.79a	16.3 ± 0.44a	30 ± 2.65c	
Park	18.2 ± 0.4a	16.9 ± 0.53a	44 ± 2.0b	
E. japonica	Road	22.7 ± 1.06b	18.6 ± 0.61b	39 ± 2.65b	
Campus	34.7 ± 1.25a	28.3 ± 0.7a	46 ± 3.46a	
Park	19.9 ± 0.66c	18.9 ± 0.36b	44 ± 3.46ab	
Note:

Different lower case letters indicate significant differences among different sampling areas for the same tree species (P < 0.05).

The photosynthetic gas exchange parameters response of five evergreen tree species to different levels of PM pollution

As shown from Fig. 4, compared with clean area (park), the photosynthetic response indexes including net photosynthetic rate (Pn), transpiration rate (E), stomatal conductance (Gs) and intercellular CO2 concentration (Ci) were decreased with pollution aggravation, with variations ranging from 7.04 to 13.64 μmol∙m−2∙s−1, 0.93 to 3.93, 0.05 to 0.14 mol∙m−2∙s−1, and 171.67 to 274.67 μmol∙mol−1, respectively. P. tobira exhibited higher values for Pn, E, and Gs compared to the other four plants.

Figure 4 The photosynthetic gas exchange parameters response of five tree species under different pollution levels.

Note: road, Zhongzhou Avenue; campus, Henan Agricultural University; park, Zhengzhou Botanical Garden. Different lowercase letters indicate significant differences among different tree species in the same sampling area (P < 0.05).

The physiological responses of five evergreen tree species to different levels of PM pollution

According to the findings in Fig. 5, it was observed that plant physiological traits exhibit regularly changes in response to atmospheric PM. The increase in atmospheric PM leads to a decrease in Chl a and Chl b content, soluble sugar and soluble protein content, as well as SOD and POD activity. Conversely, MDA content and SLW increase. The Chl a and Chl b content of five tree species varied from 0.10 to 0.77 and 0.03 to 0.26 mg·g−1, respectively. It is worth noting that in all three regions, the Chl a and Chl b content in P. serrulata was higher compared to the other four plants. Significant differences were also observed in soluble sugar (1.3–54.72%) and soluble protein content (0.48–8.72 mg·g−1) among the five tree species (P < 0.05). In addition, it was found that the soluble sugar content of E. japonica increased and then decreased with increasing pollution levels, the higher value was observed on the road, in comparison to relatively clean areas. The SOD activity ranged from 191.77–302.56 u·g−1, with the highest activity observed in E. japonicus. The POD activity (0.24–8.82 u·g−1) also differed significantly among the five tree species (P < 0.05), with P. serrulata showing the highest activity. In contrast to SOD, POD activity was more sensitive in three sampling areas. The MDA content (3.66–18.31 μmol·g−1) and SLW (92.39–181.82 mg·cm−2) exhibited the highest values on the road overall, and there were significant differences among the five plants (P < 0.05).

Figure 5 The physiological response of five tree species to different levels of PM pollution.

Different lowercase letters indicate significant differences among different tree species in the same sampling area (P < 0.05).

Fast chlorophyll fluorescence analysis of different tree species

After being irradiated with saturating pulsed light, the chlorophyll fluorescence of five dark-adapted leaves rapidly increase, stabilized after OJIP, and then slightly decreased (refer to Fig. 6). Under the stress of particulate matter pollution, the JIP values of all five plants generally declined to varying extents, resulting in inhibited photosynthesis. The most significant reduction occurred at points J and I. Among these plants, P. serrulata, E. japonicus and L. lucidum exhibited similar I values at three sampling sites. However, compared to relatively clean areas, the P value of L. lucidum decreased significantly.

Figure 6 Kinetic curves of chlorophyll fast fluorescence induction of five plants in three sampling areas.

It shows normalized by Fo and Fm to V = (Ft − Fo)/(Fm − Fo) and ΔV = Vt (treatment) − Vt (control) in a logarithmic time scale. The lowest fluorescence when exposed to light is at point O, and the highest peak of fluorescence is at point P.

Mathematical analysis was used to obtain over 50 fluorescence parameters from the fast chlorophyll fluorescence curves. Ten basic fluorescence parameters related to dust retention were selected for comparison. The botanical garden fluorescence parameters were used as the control, with a parameter value of 1. A radar map (refer to Fig. 7) was used to present the ratio of the fluorescence parameters to the control parameters of different tree species. Both TRo/RC and ETo/RC decreased with the aggravation of particle pollution, while TRo/RC showed only slight change. There was no significant difference in Fv/Fm under different levels of particle pollution. Vj, Vi, Sm, N, and Tfm all showed a downward trend with increasing particulate matter pollution level. The PSI energy utilization parameter ΦRo decreased with increasing pollution. The parameter PI(abs) represents the comprehensive performance of the light-receiving area, which also showed a decreasing trend.

Figure 7 Radar plot of chlorophyll fluorescence parameters of five plants in three sampling areas.

Correlation between PM, photosynthetic and physiological indicators of tree species

As shown from Fig. 8. correlation analysis showed that the TSP was positively correlated with PM10, PM2.5 and MDA content (P < 0.01), as well as SLW (P < 0.05). Additionally, it was negatively correlated with soluble sugar content, Pn, E, and Gs (P < 0.01), as well as Chl b, POD activity and Ci (P < 0.05). PM10 and PM2.5 were positively correlated with MDA content and SLW (P < 0.01). PM10 was negatively correlated with Chl b, soluble sugar content, POD activity, Pn, E, Ci (P < 0.01), and Chl a and Gs (P < 0.05). PM2.5 was negatively correlated with Chl b, soluble sugar content, POD activity, Pn, and E (P < 0.05), as well as Ci (P < 0.01). There was no significant correltaion between the soluble protein content, SOD activity, and dust retention. Among them, the response of MDA to atmospheric particulate matter was the most severe, with Pearson correlation coefficient value of 0.65.

Figure 8 Correlation between leaf dust retention, photosynthetic and physiological traits of five evergreen tree species.

Note: Chla, chlorophyll a; Chlb, chlorophyll b; SS, soluble sugar content; SP, soluble protein content; SOD, superoxide dismutase activity; POD, peroxidase activity; MDA, malondialdehyde content; SLW, specific leaf weight; *indicates a significant correlation (P < 0.05); **indicates an extremely significant correlation (P < 0.01).

Correlation and synergistic relationship between photosynthetic and physiological characteristics of plant leaves

Figure 9 illustrates a clear quantitative relationship between various physiological and biochemical traits under the influence of PM pollution pressure. The effect of dust treatments varied significantly among all plants’ biochemical parameters. The results revealed that MDA content was negatively correlated with chlorophyll content, osmotic regulatory substances (soluble sugar and soluble protein content) and photosynthesis indexes (including Pn, E, Gs, Ci). On the other hand, MDA content showed a positive correlation with SLW. Furthermore, SOD activity was positively correlated with Ci, and negatively correlated with other indicators.

Figure 9 Principal component analysis biplot of leaf physiological and biochemical traits.

According to Table 3, two principal components were extracted based on the criterion that the eigenvalue exceeded 1 (Specifically, the eigenvalues were 6.05 and 1.79, respectively). These two principal components accounted for 50.4% and 14.9% of the total contribution, respectively. In combination, their cumulative contribution rate was 65.3%. This suggests that these two principal components significantly influenced the variation of leaf physiological and biochemical traits. The initial factor loading matrix of the principal components (Table 3) and the loading diagram of principal component analysis (PCA) were utilized to analyse the data. The results show that chlorophyll content, osmotic regulatory substances (soluble sugar and soluble protein), POD activity and photosynthesis indices (Pn, E, Gs, Ci) were positively correlated with the first principal component (PC1). On the other hand, MDA content, SLW and SOD activity showed negative contributions. The ranking of magnitude of correlation (absolute value) is as follows: Pn > E > Gs > Chl b > Chl a > POD > Ci > SS > SLW > SP > MDA > SOD. These findings suggest that the indicators significantly related to the first principal component can serve as key indicators of leaf photosynthetic and physiological traits. The principal component contrasts Pn, E, Gs, Chlb and Chla exhibit the greatest variation.

Table 3 Factor matrix and principal component contribution rate of leaf physiological and biochemical traits.

Projects	Scores	Projects	Scores	
PC1	PC2	PC1	PC2	
Eigenvalues	6.05	1.79	Eigenvalues	6.05	1.79	
Cumulative contribution rate/%	50.4	14.9	Cumulative contribution rate/%	50.4	14.9	
Chla	0.33	0.01	MDA	−0.16	−0.05	
Chlb	0.34	0.03	SLW	−0.27	−0.12	
SS	0.27	−0.25	Pn	0.36	−0.14	
SP	0.22	−0.49	E	0.35	0.07	
SOD	−0.08	0.67	Gs	0.35	0.06	
POD	0.30	0.22	Ci	0.30	0.40	

Discussion

Comparison of PM adsorption of five tree species under different pollution levels

Using five common evergreen species in Zhengzhou as research subjects, the present study showed that the PM adsorption per unit leaf area of five plants followed the order E. japonica > E. japonicus > P. serrulata > L. lucidum > P. tobira. This indicates that E. japonica had the highest dust retention capability compared to other four species, possibly due to the presence of distinct grooves and folds on its leaf surface (Chen et al., 2017). On the other hand, the leaf surface of P. tobira was relatively smooth, resulting in the lowest PM absorption capacity. The effect of micromorphological characteristics on PM retention requires further investigation. Most of the particles trapped by plant leaves were predominantly coarse particles (PM>10), which is consistent with the findings of Dang et al. (2022). Moreover, we also observed that the increase in PM2.5 retention varies among plant species as increasing pollution, E. japonica showed an even greater increase. This suggests that the particle retention of the same plant is strongly affected by the environmental conditions, while the dust retention capacity of different tree species mainly determined by the morphological characteristics of the leaf surface (Li et al., 2023; Zhu & Xu, 2021).

The dust retention ability of plants is correlated with environmental pollution levels and pollutant sources. Previous studies have observed that the dust retention capacity of the same tree species varies in different urban functional areas, with industrial zones having the strongest dust retention capacity, followed by transport hub areas, residential areas, and clean areas (Aliya et al., 2014). In this study, the average PM (TSP, PM10, PM2.5) retention of five tree species was found to be highest on the road, followed by campus and the botanical garden, consistent with previous findings. Zhongzhou Avenue serves as a major pollution source due to exhaust emissions and the substantial dust generation. At the same time, the presence of active human activities contributes to the highest dust retention on roads (Zhu & Xu, 2021; Jeanjean, Monks & Leigh, 2016). Followed by the campus, mainly from non-motorised vehicles and infrequent human movement, resulting in relatively lower PM pollution levels in the atmosphere (Abhijith & Kumar, 2020). The botanical garden has the lowest pollution levels, due to its high vegetation coverage and the distance from the central city, there are no pollution sources nearby.

The effects of PM pollution on plant stomata, photosynthetic indexes and chlorophyll fluorescence parameters

The indicators relevant to photosynthesis, including Pn, E, Gs, and Ci, decreased with the increase of pollution, and were significantly negatively correlated with PM retention. The accumulation of particles on the leaf surface may lead to partial stomatal closure, thereby reducing the transpiration rate and carbon assimilation in photosynthesis (Khalid et al., 2018; Kumar et al., 2022). Previous studies have found that the size of plant stomata in polluted areas are smaller, and more in number, compared to control area (Kardel et al., 2010), which is consistent with the results of this study. While the stomata of most plants shrink in polluted areas, plants in these environments have developed some adaptations to assimilate more CO2 due to the increased concentrations of CO2 and other pollutants. This results in an increased number of stomata per square millimeter (Stevovic, Mikovilovic & Calic-Dragosavac, 2010; Kumar et al., 2022), which also explains the decrease in Gs, which in turn affects the physiological and biochemical processess of plants (Ghafari et al., 2021). A significant amount of dust had accumulated on the leaf surface, covering both the blade surface and the outer area, interrupting the gas exchange. This accumulation was particularly prominent on the leaf area responsible for photosynthesis, resulting in a reduced photosynthetic rate (Li et al., 2023). Additionally, the decrease in chlorophyll content disrupted the plant’s ability to carry out primary photochemical reactions, thus affecting its photosynthetic capacity (Mand et al., 2013; Sharma & Singh, 2022). This article agrees that the reduction in photosynthetic rate includes a variety of factors.

This study observed a decrease in the JIP values of the OJIP curve due to air pollution. This suggests that particulate stress inhibits the absorption and transmission of light energy in plant photosynthesis (Pourkhabbaz et al., 2010). When comparing the fluorescence parameters, it was found that PI(abs) significantly decreased with increasing pollution, indicating that particulate matter negatively affected the overall performance of the leaf photosynthetic mechanism. This hindered the progress of photochemical reactions and the accumulation of organic matter (Cuba et al., 2021). Although Fv/Fm did not show a significant difference, the results suggest that PI(abs) more accurately reflects the state of the plant photosynthetic mechanism and is more sensitive to certain stresses compared to Fv/Fm. Therefore, PI(abs) can better indicate the effect of stress on the photosynthetic mechanism (Appenroth et al., 2001; van Heerden et al., 2003; van Heerden, Strasser & Kruger, 2004). Additionally, TRo/RC and ETo/RC exhibited a decreasing trend with increasing pollution, indicating that PM pollution affected the absorption, transformation, and dissipation of light energy by the plant photosynthetic organs, as well as the related loss of electron transfer (van Heerden, Kruger & Louw, 2007). Similarly, other parameters such as Vj, Vi, Sm, and N also showed a downward trend.

The physiological response of plant leaves toward PM pollution

Chlorophyll is one of the important pigments involved in photochemical reactions (Kumar et al., 2021). In the polluted areas, the five plant species had lower levels of chlorophyll compared to the control site. This is probably due to the dust accumulation on the leaf surface, which alters the microenvironment and leads to damage and degradation of chlorophyll in the leaves. As a result, the synthesis of chlorophyll is affected, ultimately causing a decrease in its content (Karmakar & Padhy, 2019). Antioxidant enzymes, such as SOD and POD, play a crucial role in protecting cells from oxidative stress (Rai, 2016). The decrease in SOD and POD enzyme activity suggests that dust pollution stress may inhibit their synthesis in leaves. Consequently, the oxygen radicals produced by the plant exceed the scavenging capacity of the antioxidant enzyme system, leading to damage to the structure and function of the membrane system (Chaudhary & Rathore, 2019). In addition, there was a general increase in MDA content as increasing pollution levels in all five plants. This can be attributed to the higher level of damage to the cell membrane caused by the plants’ exposure to more severe road pollution (Lyu et al., 2023). However, no significant correlation was found between SOD activity and TSP, PM10, and PM2.5 retention per unit leaf area (P > 0.05), which differs from the results reported by Gao (2016), suggesting that it may not be susceptible to the stress caused by dust accumulation.

The soluble sugar content of the five plants showed different trends, and the soluble sugar content of E. japonica showed a trend of first increasing and then decreasing. This suggests that mild stress can stimulate the plant’s defence response mechanism to remove excess free radicals, giving the plant the ability to resist stress. When dust stress exceeds the tolerance limit of plants, membrane lipid peroxidation is exacerbated, normal cell metabolism is disrupted, protein synthesis is inhibited, and plant growth is impaired (Lee Hee & Lee Bum, 2000; Yin et al., 2023). Additionally, the decrease in leaf soluble sugar content may be associated with the inhibition of photosynthesis (Tzvetkova & Kolarov, 1996). We therefore speculate that this may also be the reason for the decrease in soluble sugar content of the leaves. Further research has shown that there is a positive correlation between specific leaf weight (SLW) and leaf surface dust retention (P < 0.05). Specific leaf weight (SLW) refers to the dry mass of leaves per unit area. In this study, we found that the relatively high SLW of urban plants in high pollution areas resulted from their long-term adaptation to urban air particulate pollution (Zhu & Xu, 2021).

Conclusion

This study investigated the physiological and photosynthetic responses of plants under PM pollution. The capacity of the five evergreen plants followed the order E. japonica > E. japonicus > P. serrulata > L. lucidum > P. tobira, and the difference in adsorption ability of different tree species were mainly influenced by the different morphological and structural characteristics of the leaf surface. PM pollution negatively affected the photosynthetic rate of plants as well as the overall performance of the leaf photosynthetic mechanism, P. tobira has the highest tolerance to PM pollution. Further, under appropriate levels of PM pollution, plants actively regulated osmotic pressure and scavenge excess free radicals to maintain normal growth, and when the plant’s tolerance limit is exceeded, antioxidant enzyme activity (SOD, POD) decreases, causing damage to plants. However, plants have limited resources, and investing more in one trait inevitably reduces investment in other indicators. Our findings provide valuable insights into the selection of PM-tolerant tree species. In addition, the study areas and tree species conducted in this study were limited, and further research is needed in more areas to consider human health and plant growth.

Supplemental Information

Supplemental Information 1 Raw data.

Additional Information and Declarations

Competing Interests

Author Contributions

Field Study Permissions

Data Availability

The authors declare that they have no competing interests.

Dan He conceived and designed the experiments, analyzed the data, prepared figures and/or tables, authored or reviewed drafts of the article, and approved the final draft.

Jiangqin Yuan performed the experiments, analyzed the data, prepared figures and/or tables, authored or reviewed drafts of the article, and approved the final draft.

Runze Lin performed the experiments, analyzed the data, prepared figures and/or tables, and approved the final draft.

Dongbo Xie performed the experiments, analyzed the data, prepared figures and/or tables, and approved the final draft.

Yifei Wang performed the experiments, analyzed the data, prepared figures and/or tables, and approved the final draft.

Gunwoo Kim conceived and designed the experiments, authored or reviewed drafts of the article, and approved the final draft.

Yakai Lei conceived and designed the experiments, authored or reviewed drafts of the article, and approved the final draft.

Yonghua Li conceived and designed the experiments, authored or reviewed drafts of the article, and approved the final draft.

The following information was supplied relating to field study approvals (i.e., approving body and any reference numbers):

The field experiment was approved by the Zhengzhou Botanical Garden

The following information was supplied regarding data availability:

The raw measurements are available in the Supplemental File.

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
