# Peer review of "Impact of atmospheric particulate matter retention on physiological characters of five plant species under different pollution levels in Zhengzhou"

_PeerJ, doi:10.7717/peerj.18119_

## Round 0.1 · original submission · Minor Revisions

Dear authors,

I reviewed the comments from all three reviewers, and they unanimously agreed that this manuscript is excellent, recommending only minor revisions. As an academic editor, I concur with their assessment and would like to make a decision for minor revisions. The authors have shown great potential, and with these minor adjustments, the manuscript will be even stronger.

Reviewer 1 ·

Basic reporting

- The manuscript is generally written in clear and professional English, but some sentences could be clearer.
- The introduction provides adequate context but should further explain how particulate matter affects plant physiological processes.
- The manuscript cites relevant literature, but more references on the impact of particulate matter on stomatal characteristics would strengthen the discussion.
- The structure conforms to PeerJ standards and is well-organized. Figures are relevant, high-quality, and well-labeled, though ensuring all figure legends are fully self-explanatory would be beneficial.
- Raw data is included, supporting reproducibility and transparency. Overall, the manuscript meets PeerJ's basic reporting standards, with minor improvements needed for clarity and comprehensiveness.

Experimental design

The manuscript's experimental design is sound and appropriate for the journal. There is a substantial information gap about the effect of particulate matter on plant physiology, which is addressed by the clearly stated, pertinent, and important research issue. The inquiry is thorough and upholds the highest ethical and technical standards. Enough detail is provided in the techniques to enable replication. To further improve repeatability and transparency, specific changes could include giving more information about the randomization procedure used for sample collection and making sure that every measurement approach is fully explained.

Validity of the findings

- The study, which was conducted in Zhengzhou at varying pollution levels, examines the effects of air particle matter (PM) retention on the physiological traits of five different plant species. The results are fascinating and help us understand how PM affects plant physiology, but more research is needed in a few areas before the conclusions can be considered fully accurate. One-way ANOVA and Pearson correlation analysis are used in the study to evaluate correlations and differences between different physiological and biochemical markers. Nonetheless, a thorough rationale is required for the selection of these particular statistical techniques. The manuscript should also go over the underlying assumptions of these tests and whether or not they were met. To ensure replicability, the provided raw data should be carefully examined for accuracy and clarification. This covers comprehensive metadata and dataset descriptions, unambiguous variable labeling, measurement units, and any data transformations or computations.

- The study looks at how five different plant species are affected by PM at different pollution levels, but it doesn't go far enough in addressing potential confounding variables like soil conditions and microclimates. The experimental design and data analysis should have included a discussion of how these aspects were managed or taken into consideration by the authors. Even though the publication shows statistically significant results, more should be said about the findings' biological significance, especially how long-term exposure to PM may affect the functionality and health of plants. By contrasting the results with previously published research and investigating plausible processes underlying species-specific responses, the interpretation of the data could be improved. Lastly, drawing wide conclusions from a small sample of species and places should be done with caution. It would be helpful to address the study's shortcomings in detail and recommend areas for more research. A fair assessment of the study's advantages and disadvantages should serve as the foundation for actionable suggestions for urban greening and pollution mitigation techniques. The text can greatly improve the validity and reliability of its conclusions by addressing these issues.

Additional comments

There are a few instances where the language could be improved for clarity. For example, the sentence "The results also showed that P. tobira had significantly higher Pn, E, and Gs values compared to the other four plants (P < 0.05)" could be rephrased to "P. tobira exhibited significantly higher values for Pn, E, and Gs compared to the other four plants (P < 0.05)."

·

Basic reporting

In the reviewed article titled "Impact of Atmospheric Particulate Matter Retention on Physiological Characters of Five Plant Species under Different Pollution Levels in Zhengzhou," the authors employ professional and clear English throughout their manuscript. The introduction section effectively sets the stage for the study, providing comprehensive background information and contextual relevance. It draws on a substantial body of scientific literature to support the significance of the research, emphasizing the importance of understanding how atmospheric particulate matter affects plant physiology. By integrating relevant studies, the authors successfully highlight the critical need to investigate the interactions between pollution levels and plant health in urban environments.
A weakness or limitation in the bibliography is the limited presence of articles from other parts of the world outside the country under study (China). One of the strengths of the research is certainly one of the objectives of the article, which is to investigate the effect of PM on plant physiology. This is well expressed in lines 67-69. However, in the following sentence (lines 69-70), it is unclear what the authors intend to convey. Based on that sentence, it seems that through the regulation of their physiological parameters, plants could actively mitigate PM. I believe that cause and effect have been inverted, and thus the intended meaning of this passage needs to be revised. I also do not think that Kumar et al. (2021), cited in this sentence, intended to suggest this.
The subsequent section, on the other hand, is consistent with the article and describes all the potential damage caused by air pollution to exposed vegetation. Finally, the last sentence of the introduction at line 97 could include a broader perspective: the research is certainly of interest not only for the city of Zhengzhou.
The structure of the article adheres to the standards of PeerJ, ensuring a logical and coherent presentation of the research. Additionally, the figures included in the article are relevant and of high quality. They are well-described and accompanied by appropriate captions, which enhance the reader's understanding of the data and findings presented.

Experimental design

The research clearly falls within the scope of the journal, and the research questions are well-defined and articulated in the latter part of the introduction. The study is conducted with rigor and is well-executed.
One critique regarding the methodology is the use of washing and filtration as a method for analyzing PM deposited on leaves. Although this is an established method and useful for comparing species, it is a method debated in the literature since a portion of the total suspended particles (TSP), specifically the soluble fraction, is inevitably lost during the washing process. It would be beneficial to consider this aspect, possibly including it in the discussion of the results. Relevant article to consider include is Sgrigna et al. (Sgrigna, G.; Baldacchini, C.; Dreveck, S.; Cheng, Z.; Calfapietra, C. Relationships between Air Particulate Matter Capture Efficiency and Leaf Traits in Twelve Tree Species from an Italian Urban-Industrial Environment. Sci. Total Environ. 2020, 718. https://doi.org/10.1016/j.scitotenv.2020.137310.).

Validity of the findings

The article certainly demonstrates intrinsic novelty, as highlighted in the introduction, with one of its objectives addressing a significant gap in research related to urban vegetation and/or anthropogenic stress. Furthermore, the presented data appear robust, and the differences between samples are explained through statistical significance.

However, in the sentence at lines 110-111, the specific type of particulate matter being referred to should be explicitly stated. While the reader might assume PM10, this should be clearly declared. Additionally, in line 113, the first sentence lacks a verb and is not clear at all.

Reviewer 3 ·

Basic reporting

Although the topic is of interest to the scientific community, before considering it for publication, this paper should be improved. Authors should reconsider the main objective of the paper according to the content. They should try to synthesize and emphasize the study's main findings and avoid long sentences. Furthermore, authors should avoid drawing risky conclusions.
Evaluation; Minor Revision.
1. Keywords; Must to revised; spelling and avoiding general and plural terms and multiple concepts (avoid, for example, 'and', 'of').
Unsuitable (too long) >>> Atmospheric particulate matter;
Dust retention ability
2. Abstract; The authors should be revised the abstract, it is too general. Moreover, it could be further developed, there is a lot of interesting data in the article. An informative and representative conclusion should be added to the abstract.
3. Line 48; PM10 (d < 10 µm) and PM2.5 (d < 2.5 µm) >>>> should be “PM10 (Dp ≤ 10 µm) and PM2.5 (Dp ≤ 2.5 µm)”
4. Line 141; TSP particles >>>>> should be “TSP”

Experimental design

5. The environmental parameters used in the analysis are clearly written, and it seems that a diagram showing the relationship between the parameters and the independent and dependent variables is needed.

Validity of the findings

6. In the main text, many numeric data are given with too many significant figures; 2 significant figures suffice, and 3 suffice in case the first significant figure is "1".
7. You must provide all the figures in high resolution. Make all the labels and legends more legible.
8. Conclusion; the findings could be further developed, there is a lot of interesting data in the article.

Additional comments

-

Annotated reviews are not available for download in order to protect the identity of reviewers who chose to remain anonymous.

---

## Round 0.2 · accepted · Accept

Dear Authors,

I confirm that the revisions addressed the previously minor comments. Your manuscript has been accepted in PeerJ, Congratulations!